# Structure-Based Insights into TGR5 Activation by Natural Compounds: Therapeutic Implications and Emerging Strategies for Obesity Management

**DOI:** 10.3390/biomedicines13102405

**Published:** 2025-09-30

**Authors:** Dong Oh Moon

**Affiliations:** Department of Biology Education, Daegu University, 201 Daegudae-ro, Gyeongsan-si 38453, Gyeongsangbuk-do, Republic of Korea; domoon@daegu.ac.kr; Tel./Fax: +82-53-852-6992

**Keywords:** TGR5 agonists, natural products, obesity, molecular docking

## Abstract

TGR5 has emerged as a promising therapeutic target for obesity and metabolic disorders due to its regulatory roles in energy expenditure, glucose homeostasis, thermogenesis, and gut hormone secretion. This review summarizes the structural mechanisms of TGR5 activation, focusing on orthosteric and allosteric ligand interactions, toggle switch dynamics, and G protein coupling based on cryo-EM and docking-based models. A wide range of bioactive natural compounds including oleanolic acid, curcumin, betulinic acid, ursolic acid, quinovic acid, obacunone, nomilin, and 5β-scymnol are examined for their ability to modulate TGR5 signaling and elicit favorable metabolic effects. Molecular docking simulations using CB-Dock2 and PDB ID 7BW0 revealed key interactions within the orthosteric pocket, supporting their mechanistic potential as TGR5 agonists. Emerging strategies in TGR5-directed drug development are also discussed, including gut-restricted agonism to minimize gallbladder-related side effects, biased and allosteric modulation to fine-tune signaling specificity, and AI-guided optimization of natural product scaffolds. These integrated insights provide a structural and pharmacological framework for the rational design of safe and effective TGR5-targeted therapeutics.

## 1. Introduction

G protein-coupled bile acid receptor 1 (GPBAR1), also known as Takeda G protein-coupled receptor 5 (TGR5), is a membrane-bound receptor activated by bile acids and expressed in a variety of metabolically relevant tissues, including brown and white adipose tissue, skeletal muscle, liver, intestinal L-cells, and the central nervous system [1,2,3]. Upon activation, TGR5 couples to the Gαs protein to stimulate adenylate cyclase, leading to elevated intracellular cyclic AMP (cAMP) levels and activation of downstream effectors such as protein kinase A (PKA) and cAMP response element-binding protein (CREB). These signaling cascades regulate diverse physiological processes, including energy expenditure, mitochondrial biogenesis, glucose metabolism, and appetite suppression [4,5,6].

The physiological relevance of TGR5 extends to multiple metabolic organs, positioning it as a promising therapeutic target in obesity and associated disorders. In brown adipose tissue (BAT), TGR5 activation promotes thermogenesis through the cAMP–PKA–type 2 deiodinase (D2) pathway, resulting in enhanced local triiodothyronine (T3) production and upregulation of uncoupling protein 1 (UCP1) [7,8]. In white adipose tissue (WAT), TGR5 induces browning by promoting mitochondrial fission and biogenesis, thereby converting energy-storing adipocytes into thermogenically active beige cells [9,10]. In the intestine, TGR5 stimulates glucagon-like peptide-1 (GLP-1) secretion from enteroendocrine L-cells, improving insulin sensitivity and glucose homeostasis [11,12,13]. Central TGR5 activation also modulates appetite by downregulating orexigenic neuropeptides in the hypothalamus [14,15]. Collectively, these findings highlight the potential of TGR5 agonism to address multiple pathophysiological features of obesity, including hyperphagia, insulin resistance, lipid accumulation, and reduced energy expenditure.

Despite this therapeutic promise, the clinical translation of synthetic TGR5 agonists such as INT-777 and compound 23H has been hindered by several challenges [16]. Systemic activation of TGR5 has been associated with undesirable side effects, including gallbladder filling, pruritus, and off-target metabolic responses [2,17]. In addition, species-specific differences in TGR5 pharmacodynamics complicate preclinical-to-clinical translation. These limitations have prompted the search for safer, more selective TGR5 modulators with improved pharmacological profiles. The representative synthetic TGR5 agonists are presented in Table 1.

In this context, natural products have emerged as attractive alternatives to synthetic ligands due to their structural diversity, multi-target potential, and favorable safety profiles [18,19,20]. Recent studies have identified a variety of natural compounds including oleanolic acid (OA), curcumin, betulinic acid (BA), ursolic acid (UA), and limonoids that act as TGR5 agonists and improve metabolic parameters in preclinical models of obesity. Moreover, some natural ligands exhibit gut-restricted activity, potentially avoiding systemic side effects while retaining efficacy through local TGR5 activation in intestinal tissues. These advantages suggest that natural product-based strategies may offer a safer and more physiologically nuanced approach to TGR5-targeted therapy.

This review summarizes the structural mechanisms and signaling pathways of TGR5 relevant to metabolic regulation, highlights recent advances in the identification and characterization of natural TGR5 agonists, integrates molecular docking data to explore ligand–receptor interaction patterns, and discusses future directions for the development of natural product-based TGR5 modulators.

**Table 1 biomedicines-13-02405-t001:** Representative TGR5 Agonist Candidates and Their Pharmacological Characteristics. Compounds are categorized based on their origin (e.g., BA/CDCA/CA derivatives or de novo synthetics), with emphasis on cAMP signaling, GLP-1 secretion, and dual FXR–TGR5 activity. Clinical stages are up to date as of 2025 based on PubMed and ClinicalTrials.gov records.

Compound	Type	Pharmacological Features	Clinical Stage	Ref.
INT-777	Synthetic	cAMP–PKA–D2 activation High potency & selectivity; based on cholic acid (CA) scaffold	Preclinical	[21]
18dia 2	Synthetic derivative (BA derivative)	Lipophilic 3-epi-BA derivative; potent and TGR5selective; significantly increases cAMP and GLP-1 secretion in vitro compared to parent BA	Preclinical	[22]
B1	Synthetic derivative (CA derivative)	258-fold higher TGR5 activity than CA; acts as a Positive Allosteric Modulator (PAM) via Thr131 Hbond; enhances Chenodeoxycholic Acid (CDCA) co-activation	Preclinical	[23]
MN6 (22g)	Synthetic	Developed for T2DM Regulates glucose homeostasis and insulin sensitivity	Preclinical	[24]
23(S)-m-LCA	Synthetic	Stereoselective C23-alkylation; enhances TGR5induced GLP-1 transcription	Preclinical	[25]
INT-767	Synthetic derivative (CDCA derivative)	Dual agonist (TGR5/FXR) Improved synthesis route; dual receptor targeting	Preclinical	[26]
RO5527239	Synthetic	Stimulates PYY and GLP-1 secretion via TGR5 Enhances gut hormone secretion; glucose stabilization	Preclinical	[27]
Compound **4b**	Synthetic	Strong TGR5 agonist using TMN scaffold Designed for dyslipidemia therapy	Preclinical	[28]
SB-756050	Synthetic	Increases GLP-1, potential for glucose control in T2DM First-in-human trial; tolerable, limited efficacy	Phase I/II clinical trial	[29]
BAR502	Gut-restricted synthetic	Intestinally targeted to avoid systemic side effects FXR/TGR5 dual action possible; under active development	Phase I clinical trial	[30]

## 2. TGR5-Mediated Regulatory Pathways in Obesity

### 2.1. TGR5-Mediated Thermogenesis via the cAMP–PKA–D2 Pathway

Obesity arises from a chronic imbalance between caloric intake and energy expenditure, resulting in the expansion and metabolic dysfunction of adipose tissue [31,32]. Among adipose types, BAT is thermogenically active and plays a crucial role in maintaining energy balance and glucose metabolism [33,34]. This thermogenic capacity is primarily driven by mitochondrial oxidative activity and the expression of UCP1, which dissipates the proton gradient to release energy as heat instead of producing ATP [35,36,37]. Thyroid hormone T3 is a key regulator of BAT activity [38]. Although T4 is secreted in greater amounts, T3 is the biologically active form that promotes mitochondrial turnover and induces thermogenic genes such as *UCP1*, *PRDM16*, and *PGC-1α*, along with genes involved in lipid oxidation and lipolysis [39,40,41,42].

TGR5, abundantly expressed in BAT, enhances T3 production by activating the cAMP–PKA–D2 signaling cascade. Upon binding to BAs, TGR5 stimulates the Gαs protein, which activates adenylate cyclase to produce cAMP [43,44]. Rising cAMP levels in turn activate PKA, which upregulates D2, the enzyme responsible for converting T4 into T3 [45,46,47,48]. The resulting increase in intracellular T3 enhances mitochondrial biogenesis and thermogenic gene expression, thereby promoting energy expenditure. Importantly, this mechanism is not limited to BAT; similar effects have been observed in skeletal muscle, where TGR5 and D2 are co-expressed. Therefore, the TGR5–cAMP–PKA–D2–T3 axis represents a key molecular pathway linking bile acid signaling to adaptive thermogenesis and metabolic regulation, with significant therapeutic implications for obesity and insulin resistance [49].

### 2.2. TGR5 Promotes the Browning of WAT

TGR5 contributes to the browning of WAT, a process characterized by the emergence of beige adipocytes with thermogenic properties resembling those of brown fat. TGR5 activation promotes mitochondrial fission through two coordinated mechanisms. First, it triggers ERK-dependent phosphorylation of dynamin-related protein 1 (DRP1) at Ser616, enhancing its recruitment to mitochondria where it mediates membrane constriction and division [50]. Second, TGR5 signaling activates the cAMP–PKA pathway, leading to phosphorylation of the transcription factor CREB [2,51]. Phosphorylated CREB binds to the promoter of the Mff gene and upregulates its transcription, thereby increasing MFF expression. MFF functions as a mitochondrial outer membrane receptor that recruits DRP1 to fission sites. Collectively, the ERK–DRP1 and cAMP–PKA–CREB–MFF pathways orchestrate mitochondrial fission in beige adipocytes [50], thereby contributing to mitochondrial remodeling and energy expenditure. These mechanisms complement the TGR5–cAMP–PKA–D2–UCP1 thermogenic pathway described in Section 2.1, underscoring the multifaceted role of TGR5 in promoting beige adipocyte function and energy homeostasis [52,53,54,55].

### 2.3. TGR5-Mediated Regulation of Glucose Homeostasis and Insulin Sensitivity

TGR5 activation promotes glucose homeostasis primarily through the cAMP–PKA signaling cascade, which operates distinctly in intestinal L cells and pancreatic β cells. In intestinal L cells, TGR5 activation sequentially activates cAMP, PKA, and CREB. This signaling cascade enhances the transcription of proglucagon (Gcg) and upregulates prohormone convertase 1/3 (PC1/3), which cleaves proglucagon into active GLP-1. Additionally, PKA-mediated phosphorylation promotes exocytosis of GLP-1-containing granules, thereby increasing GLP-1 secretion into the circulation [56,57,58,59].

Activation of TGR5 in pancreatic β cells enhances insulin secretion through both cAMP-dependent and Ca^2+^-linked signaling pathways [60,61,62]. When ligands such as OA or INT-777 bind to TGR5, the receptor selectively couples with Gαs proteins, which in turn activate adenylyl cyclase and elevate intracellular cAMP levels. This increase in cAMP facilitates a rise in cytosolic Ca^2+^ concentrations and activates two primary downstream effectors: protein kinase A (PKA) and exchange protein directly activated by cAMP (Epac). Studies using the cAMP analog 8-pCPT-2′-O-Me-cAMP, which selectively activates Epac without stimulating PKA, have shown that Epac activation leads to phosphoinositide (PI) hydrolysis and promotes insulin granule mobilization and exocytosis [63]. The signaling pathways regulated by TGR5 in the context of obesity are illustrated in Figure 1.

This schematic illustrates the molecular pathways activated by TGR5 in response to bile acid binding. In adipose tissue and skeletal muscle, TGR5 activation via Gαs stimulates adenylyl cyclase to increase intracellular cAMP levels. cAMP activates PKA, which enhances D2 expression and T4-to-T3 conversion, ultimately promoting the transcription of thermogenic genes such as UCP1, PRDM16, and PGC-1α in the nucleus. In white adipose tissue, TGR5 signaling promotes mitochondrial fission through the ERK–DRP1 and PKA–CREB–MFF axes, supporting beige adipocyte development. In intestinal L cells, TGR5–PKA–CREB signaling enhances proglucagon (Gcg) transcription and PC1/3 expression, resulting in increased GLP-1 synthesis and secretion. GLP-1 further stimulates insulin release from pancreatic β cells. In β cells, TGR5 activation also increases cAMP, leading to the activation of both PKA and Epac. Epac-mediated PI hydrolysis and elevated cytosolic Ca^2+^ facilitate insulin granule exocytosis. Collectively, these pathways link TGR5 activation to energy expenditure, glucose homeostasis, and improved insulin sensitivity.

## 3. Structural Mechanisms of TGR5 Activation

TGR5 contains seven transmembrane helices (TMH1–TMH7), along with three extracellular loops (ECL1–3) that contribute to ligand interaction and three intracellular loops (ICL1–3) that facilitate signal transmission to downstream effectors [64]. Recent advances in structural biology, including homology modeling and high-resolution cryo-electron microscopy (cryo-EM), have unveiled detailed mechanisms of TGR5 activation. Here, we compare and integrate the findings from three landmark studies [65,66,67], focusing on the orthosteric and allosteric binding sites, toggle switch function, and Gαs protein coupling interface.

### 3.1. Orthosteric Binding of Bile Acids and Synthetic Agonists

Macchiarulo et al. initially proposed two potential binding modes, tail to head and head to tail, for bile acids based on docking and mutagenesis. Subsequent experimental validation, particularly with compound **6** and the S270A mutant, confirmed the head to tail mode [65]. Compound **6** is a synthetic bile amine designed as a tool compound to probe ligand orientation, and its functional profile in mutagenesis assays (e.g., Tyr89 and Asn93 mutants) provided strong evidence supporting the head-to-tail binding hypothesis proposed by Macchiarulo et al. [65]. In this model, the 3αhydroxyl group of lithocholic acid (LCA) forms hydrogen bonds with N93^3.33^ in TM3, while the terminal carboxyl group interacts with S270^7.43^ in TM7. F96^3.36^ and W237^6.48^ help define a hydrophobic cavity accommodating the steroid core’s C6–C7 positions, and Y89^2.61^ stabilizes the complex via π–π stacking. These interactions collectively create a relatively shallow orthosteric pocket comprising TM2, TM3, TM6, and TM7. This region serves as the primary ligand-binding site, initiating conformational changes that propagate toward the cytoplasmic face to activate downstream signaling.

Later, Chen et al. [66] and Yang et al. [67] provided high-resolution cryo-EM structures of the TGR5–Gαs complex bound to synthetic agonists such as compound 23H and INT-777. Although the cryo-EM density for 23H was insufficient for precise modeling, mutagenesis and docking analysis suggest that synthetic agonists bind relatively deeper within the orthosteric cavity than bile acids, engaging residues in TM3, TM5, and TM6. Key interacting residues include F96^3.36^, L166^5.40^, Y240^6.51^, and S247^6.58^. Unlike bile acids, synthetic agonists induce a more substantial outward movement of TM6, facilitating G protein recruitment. The orthosteric pocket therefore not only mediates ligand specificity but also modulates the magnitude of receptor activation depending on the depth and nature of the ligand–receptor interactions.

Although the term “shallow” was not explicitly used in the original studies, it is employed here to describe the relatively surface-exposed orthosteric site engaged by bile acids, in contrast to the deeper binding mode of synthetic agonists. This distinction between the relatively shallow orthosteric binding of bile acids and the deeper engagement of synthetic agonists underscores the structural plasticity of the TGR5 ligand-binding site. Although both ligand classes occupy the same general orthosteric cavity, differences in binding depth and residue engagement suggest a gradient of ligand–receptor interaction that correlates with the extent of receptor activation. Bile acids such as LCA interact primarily with residues in TM2, TM3, TM6, and TM7 near the extracellular surface, defining a shallow binding mode. In contrast, synthetic agonists like 23H and INT-777 penetrate deeper into the transmembrane core, engaging residues in TM5 and the interior of TM6, which are associated with more pronounced conformational changes, particularly the outward displacement of TM6. These findings support a model in which the orthosteric pocket of TGR5 exhibits depth-dependent functional architecture, allowing ligands of varying chemotypes to differentially modulate receptor conformation and downstream signaling. In particular, shallow orthosteric binding of bile acids often results in weaker engagement of toggle switch residues such as W237^6.48^ and Y240^6.51^, leading to limited TM6 displacement and reduced Gαs coupling. This incomplete activation can manifest as partial agonism or biased signaling, whereas deeper synthetic agonists achieve stronger toggle switch engagement and more robust receptor activation.

### 3.2. Allosteric Site and Cooperative Modulation

Yang et al. [67] discovered a distinct allosteric site located on the intracellular-facing surface of TGR5, formed by TM3, TM4, TM5, and the second intracellular loop (ICL2). This shallow surface pocket accommodates secondary ligands such as cholesterol and 12α-hydroxylated bile acids (e.g., cholic acid). Residues L104^3.44^, L130^4.48^, and T131^4.49^ shape a hydrophobic groove suitable for allosteric ligand recognition. This site contributes to receptor regulation by enhancing the effect of orthosteric ligands through cooperative interactions. For instance, INT-777 displays enhanced efficacy when co-administered with cholic acid, an effect abolished by the T131A mutation. These results highlight the role of the allosteric site as a modulator of receptor sensitivity and provide a mechanistic basis for dual-ligand therapeutic strategies.

### 3.3. Toggle Switch Mechanism

The toggle switch, a hallmark microswitch in class A GPCRs, is crucial for signal transduction. In TGR5, this switch is composed of W237^6.48^ and Y240^6.51^ within TM6. W237 is involved in the initial structural rearrangement upon bile acid binding [65], while cryo-EM studies have shown that Y240 undergoes rotational movement upon synthetic agonist engagement, stabilizing the active, outward-facing conformation of TM6 [66,67]. These residues act as conformational conduits that couple ligand binding to intracellular activation. Mutation of Y240 to alanine disrupts this process, abolishing receptor function and underscoring the central role of the toggle switch in facilitating structural transitions from the ligand-binding pocket to the G protein interface.

### 3.4. Gαs Binding Interface

Activation of TGR5 leads to substantial structural remodeling on the intracellular side, enabling interaction with the stimulatory G protein Gαs. Cryo-EM studies [66,67] show that TM6 shifts outward by approximately 9 Å, creating a cavity for the insertion of the C-terminal α5-helix of Gαs. This interface is composed of residues from TM3 (E109^3.49^), TM5 (V178^5.32^, V188^5.62^, Q195^5.69^), TM6, and a positively charged motif in ICL3 (R201–R208). This surface operates as the effector coupling site, converting ligand-induced structural rearrangements into downstream cAMP signaling. The observed interaction pattern is consistent with other Gαs-coupled receptors, yet distinct in its reliance on specific ICL3 charge interactions that stabilize the TGR5–G protein complex. The structure and functional sites of TGR5 are presented in Figure 2 and Table 2.

Three-dimensional representation of the human TGR5 receptor (PDB ID: 7BW0) showing critical regions involved in ligand binding and receptor activation. The seven transmembrane helices (TM1–TM7) are shown in red, while extracellular and intracellular loops are in grey or green. The shallow orthosteric site (magenta box) accommodates bile acids and includes residues such as N93^3.33^, F96^3.36^, W237^6.48^, and S270^7.43^. The canonical orthosteric site (blue box) for synthetic agonists (e.g., INT-777, 23H) is formed by residues including L166^5.40^, S247^6.58^, F96^3.36^, and Y240^6.51^. The allosteric site (green box), located at the intracellular-facing surface, includes L104^3.44^, L130^4.48^, and T131^4.49^ and binds modulators such as cholesterol and cholic acid. The toggle switch, composed of W237^6.48^ and Y240^6.51^ (yellow labels), transduces ligand binding into TM6 movement, facilitating receptor activation. The Gαs proteinbinding interface (black box) on the intracellular side involves R201, R204, and R208 from ICL3 and accommodates the α5-helix of the G protein. This structural model illustrates the cooperative mechanism through which different ligands engage distinct sites to modulate TGR5 signaling.

## 4. Natural TGR5 Agonists: Metabolic Benefits and Mechanistic Insights

Although synthetic agonists such as INT-777 and compound 23H exhibit high potency in activating TGR5, their clinical translation is hindered by adverse effects, notably gallbladder filling and pruritus. In contrast, natural products offer structural diversity and generally favorable safety profiles, positioning them as attractive alternatives or complements to synthetic ligands.

### 4.1. Oleanolic Acid (OA)

OA, a natural pentacyclic triterpenoid found in various plant-based foods and olive leaves, has recently emerged as a bioactive compound with therapeutic potential against obesity. A growing body of evidence demonstrates that OA exerts its metabolic benefits in part through activation of TGR5, a bile acid-sensing receptor involved in energy expenditure, appetite control, and glucose homeostasis.

Recent findings show that OA significantly reduces food intake by activating TGR5-dependent cAMP signaling in the hypothalamus. TGR5 activation by OA increases intracellular cAMP levels, leading to the suppression of orexigenic neuropeptides such as NPY and AgRP. This effect was confirmed by both behavioral assays and molecular analysis of hypothalamic signaling pathways, suggesting that OA’s anorexigenic effect is mechanistically dependent on central TGR5 activation [15].

In skeletal muscle, OA has been shown to enhance slow-twitch (oxidative) muscle fiber formation, thereby improving mitochondrial oxidative capacity. This metabolic shift is mediated through the TGR5–CaN (calcineurin) signaling pathway, which activates transcription factors involved in mitochondrial biogenesis and muscle differentiation. By promoting the oxidative phenotype, OA enhances energy utilization and contributes to systemic metabolic improvement in the context of obesity [68].

Beyond appetite and muscle remodeling, OA also acts on pancreatic β-cells. Activation of TGR5 by OA stimulates insulin secretion via a PKA-dependent pathway, reinforcing its role in glucose homeostasis. This effect was abolished by pharmacological inhibitors of Gαs and adenylyl cyclase, confirming the specificity of the TGR5–cAMP–PKA axis in mediating β-cell responsiveness to nutrient signals [69].

In a diet-induced obesity mouse model, OA-rich olive leaf extract prevented weight gain, improved exercise capacity, and mitigated cognitive decline. These systemic benefits were attributed to the combined antioxidant and TGR5 agonist activity of OA. Notably, TGR5 activation was associated with increased thermogenesis and lipid utilization, positioning OA as a dual-action molecule for both metabolic and neurobehavioral disorders linked to obesity [70].

### 4.2. Curcumin

Accumulating evidence from preclinical and clinical studies suggests that curcumin modulates metabolic homeostasis by influencing the gut microbiota and bile acid receptor signaling, particularly through TGR5.

In murine models of high-fat diet-induced obesity, curcumin supplementation reduced body weight gain and improved cold tolerance by activating UCP1-dependent thermogenesis, consistent with the TGR5–cAMP–PKA–UCP1 axis described in Section 2.1. These thermogenic benefits were abolished in UCP1- and TGR5-deficient mice, confirming the requirement of this pathway [71].

Building on this, a separate study in ob/ob mice demonstrated that curcumin functions as a natural TGR5 agonist and FXR antagonist. By reshaping gut microbiota and bile acid profiles, curcumin increased deoxycholic acid (DCA) levels and promoted intestinal L-cell expansion, thereby enhancing GLP-1 secretion, in line with the mechanism detailed in Section 2.3. Importantly, curcumin simultaneously activated TGR5 and repressed FXR signaling in intestinal tissues, leading to improvements in glucose homeostasis and energy expenditure [72].

These mechanistic insights were extended to humans in a 24-week randomized controlled trial in patients with nonalcoholic simple fatty liver disease (NASFL). Curcumin (500 mg/day) significantly reduced hepatic fat content, body weight, and insulin resistance markers. Gut microbiota analysis revealed increased Bacteroides abundance and a reduced Firmicutes/Bacteroidetes ratio. Notably, circulating DCA levels and TGR5 expression in peripheral blood mononuclear cells were significantly elevated, alongside enhanced GLP-1 secretion, supporting the activation of the gut–bile acid–TGR5–GLP-1 axis in humans [73].

### 4.3. Betulinic Acid (BA)

BA, a natural pentacyclic triterpenoid, has attracted significant attention for its pharmacological potential as a selective agonist of the TGR5.

The earliest systematic evaluation of BA as a TGR5 agonist was conducted through a structure–activity relationship (SAR) study that compared the TGR5 agonist potential of BA, oleanolic acid, and ursolic acid. BA exhibited strong TGR5 activation with selectivity over FXR, another bile acid receptor. Importantly, BA induced GLP-1 secretion in intestinal cells, implicating its role in promoting satiety and glycemic control [22].

To improve the efficacy of BA as a therapeutic ligand, a series of highly lipophilic 3-epi-betulinic acid derivatives were synthesized. These compounds demonstrated increased cellular potency and maintained selectivity for TGR5 over other bile acid receptors. This chemical optimization highlighted BA’s potential as a lead scaffold for designing metabolically beneficial TGR5-targeting drugs [74].

Using NCI-H716 enteroendocrine cells that express TGR5 endogenously, it was confirmed that BA significantly enhanced GLP-1 secretion via TGR5 activation. This study provided strong functional evidence that BA stimulates gut hormone release, supporting its role in appetite regulation and glucose homeostasis, two central mechanisms in obesity management [75].

Further refinement led to the identification of BA derivatives with high agonist activity in human and canine TGR5, but not in murine TGR5 due to species-specific differences. In humanized TGR5(H88Y) knock-in mice, these derivatives improved glucose tolerance and insulin sensitivity, reinforcing the translational relevance of BA analogs for human metabolic disorders [76].

The most recent advancement focused on overcoming systemic side effects associated with TGR5 agonism, such as gallbladder filling. By modifying BA’s physicochemical properties, researchers developed gut-restricted BA derivatives that retain strong local TGR5 activity in the intestine without systemic exposure. One lead compound, 22-Na, showed favorable metabolic effects while minimizing adverse events, paving the way for safer therapeutic applications in obesity and metabolic syndrome [77]. These observations underscore the species-specific pharmacology of TGR5 agonists, as BA derivatives exhibit strong efficacy in human and canine receptors but only weak activity in murine TGR5. Such interspecies differences highlight the limitations of relying solely on rodent models for drug development and emphasize the value of humanized TGR5 mice (e.g., TGR5^H88Y) or human-derived cellular systems to improve translational relevance.

### 4.4. Ursolic Acid (UA)

UA, a naturally occurring triterpenoid, was found to act as a selective agonist of the bile acid receptor TGR5 in a study using type 1-like diabetic rats [78]. The activation of TGR5 by ursolic acid significantly increased intracellular cAMP levels in intestinal L-cells, which led to enhanced secretion of GLP-1. Since GLP-1 plays a central role in glucose-dependent insulin secretion and appetite regulation, this pathway may offer metabolic benefits beyond glycemic control. Importantly, the effects of UA on GLP-1 release were abolished by TGR5 inhibitors, confirming the specificity of the TGR5-mediated signaling cascade. Although the study was conducted in a diabetic model, the findings suggest a potential application in obesity management, as GLP-1 is known to influence energy balance and food intake. Thus, UA may contribute to metabolic improvement through the TGR5–GLP-1 axis, representing a promising candidate for further research in metabolic disease therapy.

### 4.5. 5β-Scymnol and 5β-Scymnol Sulfate

The study by Halkias et al. [79] identified two marine-derived bile compounds, 5β-scymnol and 5β-scymnol sulfate, as novel TGR5 agonists. Using HEK293 cells overexpressing TGR5, both compounds significantly increased intracellular calcium (Ca^2+^) levels in a TGR5- and Gαq-dependent manner. In contrast, mammalian bile acids like deoxycholic acid (DCA) and ursodeoxycholic acid (UDCA) showed either non-specific or no activation of TGR5. This is notable because endogenous bile acids are weak TGR5 agonists, limiting their use in therapy. In contrast, scymnol derivatives from marine sources provided strong and selective TGR5 activation. Although the study focused on atherosclerosis, TGR5 also regulates energy metabolism, thermogenesis, and glucose homeostasis, suggesting potential relevance for obesity treatment. These compounds may therefore serve as promising leads for developing marine-derived therapeutics targeting TGR5 in metabolic diseases.

### 4.6. Quinovic Acid (QA)

QA and its derivatives, isolated from the traditional medicinal plant Fagonia cretica, act as direct TGR5 agonists, significantly enhancing GLP-1 secretion and gene expression in STC-1 intestinal endocrine cells [80]. Ethyl acetate partitioning enriched GLP-1 activity, and bioassay-guided fractionation identified three active compounds: QA and two derivatives, QA-3β-O-β-D-glycopyranoside and QA-3β-O-β-D-glucopyranosyl-(28 → 1)-β-D-glucopyranosyl ester, with greater potency than QA in stimulating GLP-1 release. Mechanistically, QA-activated TGR5 signaling increased intracellular GLP-1 and upregulated proglucagon, GIP, and PC1/3 expression. These findings highlight QA and its derivatives as promising natural incretin-enhancing agents, with potential applications in obesity and type 2 diabetes via gut hormone modulation.

### 4.7. Obacunon

Horiba et al. investigated obacunone in KKAy obese mice. Dietary supplementation of obacunone led to reduced body weight, decreased adiposity, and improved glycemic control [81]. Mechanistically, obacunone enhanced the transcriptional activity of both TGR5 and PPARγ, suggesting that it exerts its effects through dual regulation of energy expenditure and lipid/glucose metabolism. It also upregulated TGR5 mRNA in skeletal muscle, contributing to muscle hypertrophy and improved metabolic profile.

### 4.8. Nomilin

Ono et al. explored the effects of nomilin, another citrus limonoid, in high-fat diet-fed mice. Nomilin treatment significantly suppressed body weight gain and fasting glucose levels [59]. Using TGR5 reporter assays, nomilin was confirmed as a TGR5 agonist, showing direct activation comparable to known agonists. These metabolic benefits were associated with increased cAMP signaling, a downstream effector of TGR5.

Curcumin has demonstrated clinical efficacy in a 24-week randomized controlled trial in patients with nonalcoholic simple fatty liver disease, where supplementation reduced hepatic fat content, body weight, and insulin resistance while elevating circulating DCA levels, TGR5 expression, and GLP-1 secretion. By contrast, other natural TGR5 agonists such as betulinic acid, ursolic acid, and quinovic acid have shown promising metabolic effects in preclinical models but lack clinical validation in humans, leaving their translational potential uncertain. To bridge this gap, future research should prioritize advancing these natural compounds into well-designed clinical trials to evaluate their safety, bioavailability, and therapeutic efficacy, thereby strengthening the clinical relevance of TGR5-targeted natural products. Natural compounds regulating TGR5 activity and metabolism are depicted in Figure 3 and Table 3.

Schematic representation of natural products that activate TGR5 and modulate key metabolic processes relevant to obesity and related disorders. Each compound is shown with its chemical structure and primary metabolic effects, including GLP-1 secretion, thermogenesis, appetite suppression, glucose homeostasis, and energy expenditure. The central TGR5 receptor diagram highlights the orthosteric and shallow orthosteric binding sites. Compounds such as oleanolic acid, betulinic acid, ursolic acid, curcumin, 5β-scymnol, nomilin, obacunone, and quinovic acid engage TGR5 via diverse mechanisms and signaling pathways (e.g., cAMP, PKA, PPARγ, and GLP-1 axis), supporting their potential as therapeutic candidates for metabolic diseases.

## 5. Natural Product Binding to TGR5: Docking Insights into Orthosteric and Allosteric Modulation

Recent structural biology advances, particularly cryo-EM and molecular modeling, have elucidated the mechanisms of TGR5 activation, underscoring the distinct roles of orthosteric and allosteric sites, as well as key microswitch residues like W237^6.48^ and Y240^6.51^ that govern G protein engagement. To explore the potential of natural products as TGR5 modulators, a molecular docking analysis was conducted on a panel of bioactive compounds, including OA, curcumin, BA, UA, 5β-scymnol, QA, obacunone, and nomilin. These ligands demonstrated varying binding affinities and interaction patterns within the orthosteric or shallow orthosteric sites of TGR5. Docking simulations were performed using the CB-Dock2 platform, which has been benchmarked across diverse protein–ligand systems and reported to achieve an ~85% success rate for binding pose prediction (RMSD < 2.0 Å) [82], supporting its suitability for generating interaction hypotheses in this study.

OA and curcumin both exhibited binding scores of −8.8 kcal/mol and occupied the canonical orthosteric pocket defined by residues in TM3 and TM6. OA formed conventional hydrogen bonds with Leu71^1.49^ and Leu74^2.66^, and engaged in van der Waals interactions with key residues such as Asn93 (N93^3.33^), Ser247 (S247^6.58^), Tyr89^2.61^, and Val248^6.59^. Curcumin, on the other hand, established hydrogen bonds directly with Asn93^3.33^ and Ser247^6.58^, and exhibited hydrophobic alkyl and π–alkyl interactions with Leu71, Leu246, Leu265, and Leu266. These interactions suggest favorable accommodation within the orthosteric pocket, potentially stabilizing an active-like receptor conformation by engaging residues associated with bile acid and synthetic agonist recognition.

In contrast, BA (−7.3 kcal/mol) exhibited a notably weaker binding affinity, while UA (−9.0 kcal/mol) showed stronger binding. Both BA and UA occupied a more superficial region of the orthosteric cavity, consistent with the “shallow orthosteric site” described in recent cryo-EM studies [65,66,67]. BA formed hydrogen bonds with Ser156 and Ser157 and engaged in π–alkyl and van der Waals interactions with residues such as Trp149, Tyr251^7.31^, and Glu252, whereas UA established hydrogen bonds with Val248 and Trp75 and interacted via van der Waals forces with Leu71^1.49^ and Tyr89^2.61^. The relatively superficial engagement of BA and UA, lacking strong contacts with toggle switch residues such as W237^6.48^ or Y240^6.51^, implies reduced capacity to induce the conformational changes necessary for full TM6 displacement and efficient Gαs recruitment.

5β-scymnol also targeted the shallow orthosteric site, with a binding score of −8.7 kcal/mol. It formed multiple hydrogen bonds with N93^3.33^, S247^6.58^, and Y251^7.31^, as well as π–alkyl interactions with W75 and Y89. The simultaneous engagement of polar and aromatic residues may stabilize an intermediate receptor conformation, suggesting the possibility of partial or biased agonist behavior. Quinovic acid, with a binding score of −9.1 kcal/mol, similarly engaged N93^3.33^ and L74 while occupying a position adjacent to F96^3.36^, a residue implicated in synthetic agonist binding. Such interactions underscore the potential for modulating TGR5 activation from an intermediate orthosteric position.

Obacunone and nomilin demonstrated the strongest binding affinities (−9.9 and −9.7 kcal/mol, respectively), driven by hydrogen bonds to N93^3.33^ and Y251^7.31^, π–alkyl interactions with L71^1.49^, and van der Waals contacts with L166^5.40^ and S247^6.58^. These compounds showed extensive interaction networks within the shallow orthosteric region, overlapping with the binding zones of both bile acids and synthetic agonists. Notably, concurrent interactions with residues involved in both shallow and deeper orthosteric regions, including F96^3.36^ and Y251^7.31^, suggest potential for orthosteric-allosteric cooperativity, aligning with the dual-ligand modulation paradigm described by Yang et al. [67].

Collectively, these findings demonstrate that natural products predominantly interact with the orthosteric or shallow orthosteric pocket of TGR5, leveraging critical residues such as N93^3.33^, S247^6.58^, and Y251^7.31^. Although none of the tested compounds directly occupied the intracellular allosteric site involving TM4 and ICL2 [67], their varied interaction depths and contact profiles imply differential capacities for toggle switch engagement and Gαs coupling. This supports the notion of ligand-specific activation, where binding depth and orientation modulate the extent of TM6 displacement and downstream signaling efficacy. Future structure–activity relationship (SAR) studies and functional assays are warranted to evaluate whether these docking patterns correlate with partial, biased, or full agonist activity in cellular contexts. Figure 4 and Table 4 show docking-based interaction profiles of selected natural products with TGR5. It should be emphasized that docking results are hypothesis-generating rather than definitive evidence. These computational approaches have inherent limitations, as they cannot fully capture receptor conformational flexibility, membrane context, solvation effects, or binding kinetics. Thus, docking data should be interpreted with caution and ideally corroborated by biochemical or structural validation.

## 6. Emerging Strategies for TGR5-Directed Therapeutics

Despite substantial preclinical evidence that TGR5 agonists improve metabolic health and glucose homeostasis, systemic administration of these ligands often leads to adverse events such as gallbladder filling and pruritus, limiting translational potential. To overcome these limitations, several forward-thinking strategies have emerged to enhance safety and efficacy.

### 6.1. Gut-Restricted TGR5 Agonists

Designing gut-restricted TGR5 agonists represents a major innovation. These compounds are structurally modified to remain localized in the intestine, thereby avoiding systemic exposure and associated gallbladder side effects while retaining metabolic efficacy. Notably, a gut-restricted derivative of betulinic acid, termed 22-Na, showed robust TGR5 activation in humanized TGR5^H88Y mice and significantly improved glucose tolerance without causing gallbladder filling [77].

### 6.2. Allosteric and Biased Agonism

Structural studies have identified a putative allosteric site on TGR5 (formed by TM3–TM5 and ICL2), opening opportunities for allosteric modulation that fine-tunes receptor responsiveness [67]. Ligands capable of biased agonism, selectively enhancing branches of signaling such as GLP-1 secretion while minimizing undesired effects like gallbladder stimulation, are another promising direction. For instance, compounds engaging Thr131 as a positive allosteric modulator (PAM) exhibit synergistic activation with endogenous bile acids and support the paradigm of dual-ligand activation. While structural studies have revealed a potential allosteric site in TGR5, no natural products have yet been experimentally validated as allosteric modulators. Therefore, this concept should be regarded as a promising avenue for future research rather than established evidence, with opportunities to identify ligands that fine-tune receptor activity beyond the classical orthosteric mechanism.

### 6.3. AI-Guided Optimization of Natural Product Scaffolds

Recent advances in artificial intelligence (AI) and deep learning have enabled high-throughput virtual screening and scaffold optimization for TGR5 agonists. AI-guided docking and structure–activity relationship (SAR) modeling can rapidly evaluate large libraries of natural products. For example, machine learning–driven molecular modeling has predicted strong binding of compounds such as obacunone, nomilin, and quinovic acid to the TGR5 orthosteric pocket. These approaches facilitate structure-based refinement, aiming toward properties such as gut restriction, signaling bias, and improved selectivity. These emerging strategies collectively offer a multi-dimensional framework for developing safer and more effective TGR5-targeted therapies, integrating chemical design, receptor pharmacology, and computational innovation.

## 7. Conclusions

TGR5 represents a compelling molecular target for addressing the multifactorial pathogenesis of obesity, encompassing energy imbalance, insulin resistance, and dysregulated gut hormone signaling. Its structural versatility enables differential ligand engagement within orthosteric and allosteric domains, modulating downstream signaling intensity and specificity.

Natural compounds such as OA, BA, curcumin, and QC demonstrate potent TGR5 agonism and favorable metabolic effects with lower risk of systemic toxicity. These ligands frequently promote GLP-1 secretion, browning of white adipose tissue, and enhanced thermogenesis. Notably, gut-restricted derivatives of BA (e.g., 22-Na) maintain metabolic efficacy while avoiding gallbladder filling, a major limitation of systemic TGR5 agonists. Despite promising preclinical efficacy, natural TGR5 agonists face significant translational challenges, including poor oral bioavailability (e.g., curcumin), potential side effects such as gallbladder filling observed with systemic agonists, and variability in plant extract preparations that complicates reproducibility and standardization.

Beyond traditional orthosteric binding, biased and allosteric TGR5 agonists offer a promising therapeutic approach by enabling pathway-selective modulation. Such ligands may preferentially activate cAMP-PKA or calcium-dependent cascades, maximizing metabolic efficacy while minimizing off-target effects.

In summary, integrating structure-guided drug design, gut-restricted ligand delivery, and biased signaling concepts offers a refined framework for the development of safe and effective TGR5-directed therapeutics for obesity and related metabolic disorders.

## Figures and Tables

**Figure 1 biomedicines-13-02405-f001:**
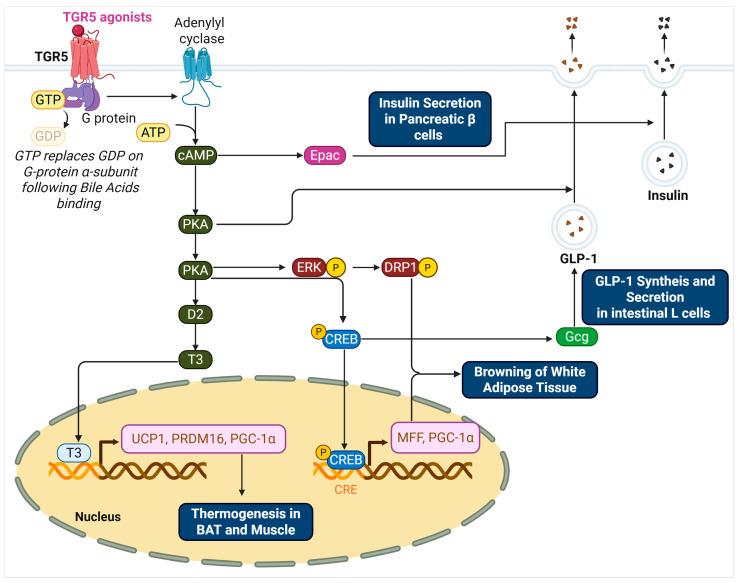
TGR5-Mediated Signaling Pathways Regulating Thermogenesis, Browning, and Insulin Secretion.

**Figure 2 biomedicines-13-02405-f002:**
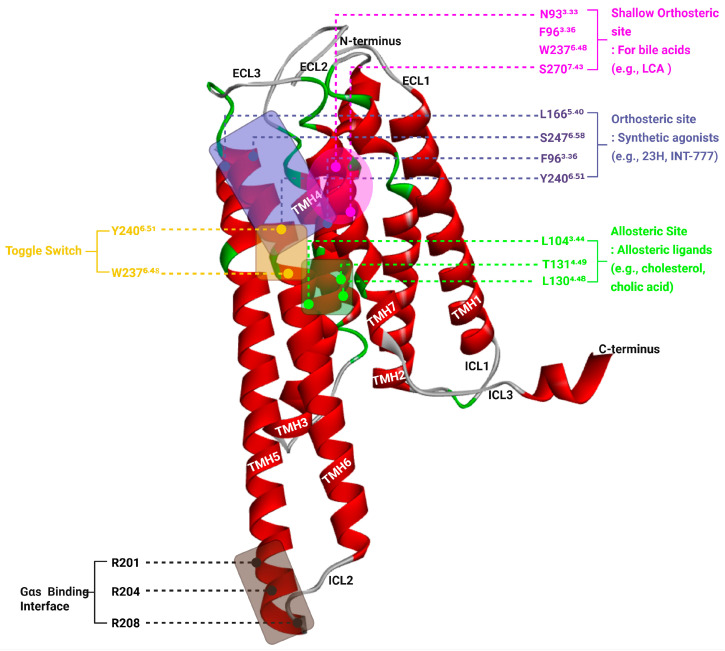
Structural Features of TGR5 Highlighting Orthosteric, Allosteric, Toggle, and Gαs Coupling Sites.

**Figure 3 biomedicines-13-02405-f003:**
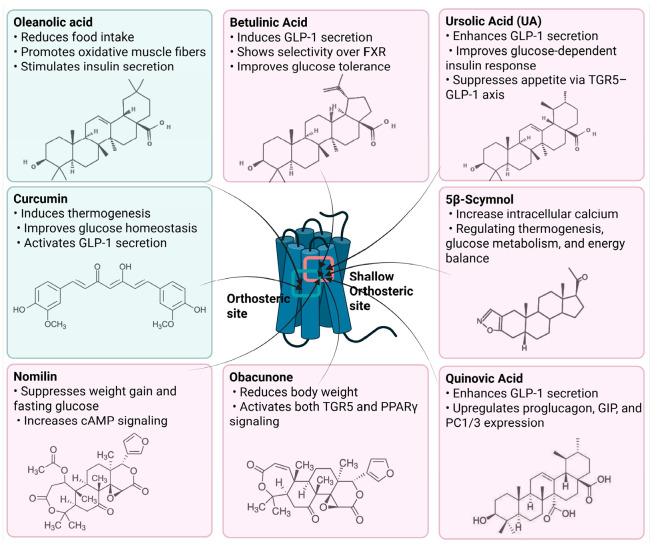
Natural Compounds Targeting TGR5 and Their Metabolic Effects.

**Figure 4 biomedicines-13-02405-f004:**
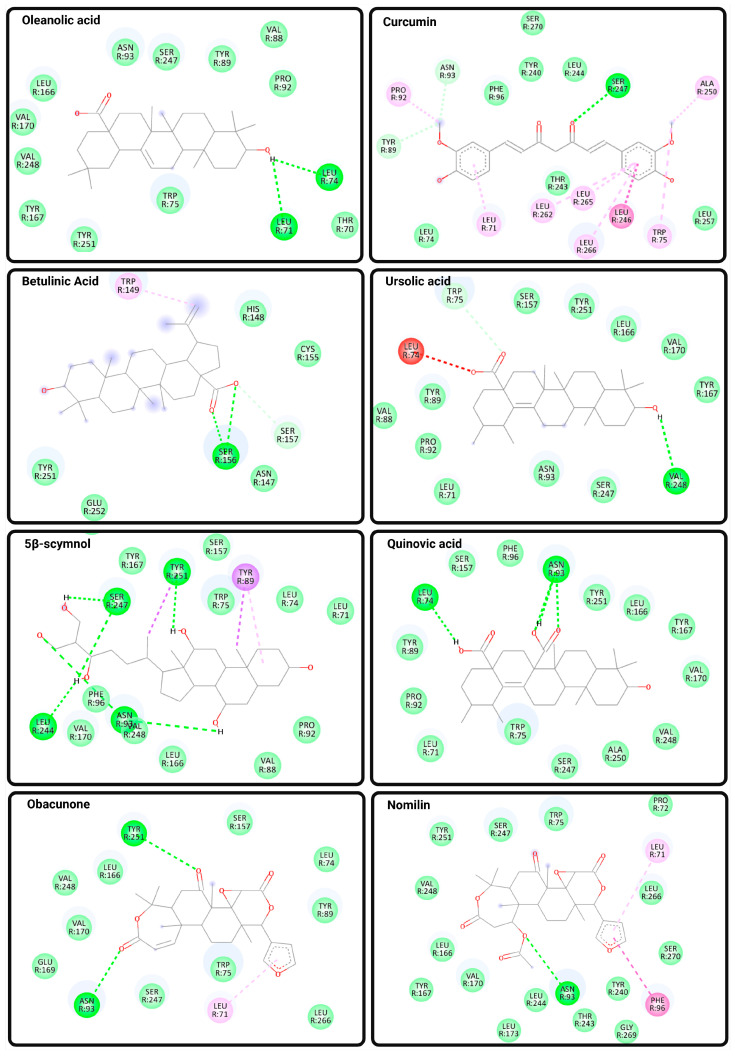
Predicted Binding Interactions Between Natural Compounds and TGR5. Molecular docking poses and binding interactions of eight natural compounds with the bile acid receptor TGR5 (PDB ID: 7BW0) were visualized using Discovery Studio Visualizer. Docking was performed using CB-Dock2, and ligand structures were obtained from PubChem in SDF format. Key binding interactions include conventional hydrogen bonds (bright green dashed lines), carbon hydrogen bonds (light green lines), and van der Waals interactions (light green shading), as well as hydrophobic contacts, including π–alkyl, alkyl, and amide–π stacking interactions (shaded pink and purple lines). The selected natural ligands include oleanolic acid, curcumin, betulinic acid, ursolic acid, 5β-scymnol, quinovic acid, obacunone, and nomilin. These compounds exhibit distinct binding profiles within the orthosteric region of TGR5, supporting their potential roles as modulators of TGR5 activity.

**Table 2 biomedicines-13-02405-t002:** Functional Sites of TGR5. Table summarizes the structural locations and functional roles of key activation sites in TGR5, including the orthosteric site, allosteric site, toggle switch, and Gαs binding interface, based on cryo-EM structures and mutational analyses.

Site	Key Residues	Function	Ligand Type	Structural Features	Ref.
Orthosteric Site	Y89^2.61^, N93^3.33^, F96^3.36^, L166^5.40^, Y240^6.51^, S247^6.58^, S270^7.43^	Primary ligand-binding site; initiates conformational changes leading to G protein activation. Binding depth and contact profile determine activation strength.	Bile acids (e.g., LCA), Synthetic agonists (e.g., INT-777, 23H)	For bile acids: The orthosteric pocket is relatively shallow and located near the extracellular surface, formed by TM2, TM3, TM6, and TM7. N93^3.33^ and S270^7.43^ form polar interactions with the 3α-OH and carboxyl group. F96^3.36^ and W237^6.48^ form a hydrophobic groove. For synthetic ligands: INT-777 and 23H bind deeper within the TM core, engaging TM3, TM5, and TM6. F96^3.36^, L166^5.40^, Y240^6.51^, and S247^6.58^ form a deep hydrophobic pocket that stabilizes the ligand and promotes TM6 displacement.	[65,66,67]
Allosteric Site	L104^3.44^, L130^4.48^, T131^4.49^	Enhances orthosteric ligand efficacy via cooperative binding; modulates receptor sensitivity.	Cholesterol, 12α-OH BAs (e.g., cholic acid)	Shallow surface groove between TM3–TM4–TM5–ICL2; T131A abolishes synergy.	[67]
Toggle Switch	W237^6.48^, Y240^6.51^	Couples ligand binding to TM6 movement; essential for activation signal propagation.	All orthosteric ligands	Rotational shift of Y240 stabilizes outward TM6; W237 engages C6 substituent of bile acids.	[67]
Gαs Binding Interface	E109^3.49^, V188^5.62^, Q195^5.69^, V178^5.32^, R201, R204, R208	Mediates Gαs coupling and downstream cAMP signaling via electrostatic and hydrophobic interactions.	Gαs protein	TM6 outward shift creates binding cleft for Gαs α5-helix; stabilized by ICL3 charge motif.	[66,67]

**Table 3 biomedicines-13-02405-t003:** Overview of Natural Products That Modulate TGR5 Signaling and Metabolic Outcomes.

Compounds	Study Model	Key Mechanisms Involved	TGR5-Related Findings	Metabolic Outcomes	Ref.
Oleanolic acid (OA)	Mouse (HFD-fed)	OA activates TGR5–cAMP signaling in the hypothalamus to regulate appetite	Central TGR5 activation increases cAMP, downregulates orexigenic peptides	↓ Food intake, ↓ body weight	[15]
	Skeletal muscle study	OA promotes slow-twitch muscle fiber formation via TGR5–CaN signaling	TGR5 activation enhances oxidative metabolism through calcineurin–MEF2/PGC-1α axis	↑ Mitochondrial function, ↑ energy expenditure	[68]
	Isolated β-cells	OA stimulates insulin secretion through cAMP–PKA pathway	TGR5 activation in β-cells promotes insulin release; blocked by AC or Gαs inhibitors	↑ Insulin secretion, improved glucose homeostasis	[69]
	Obese mice (DIO model)	OA-rich olive leaf extract improves systemic metabolism via TGR5 activation	OA activates TGR5 systemically; enhances lipid utilization and thermogenesis	↓ Body weight, ↑ exercise capacity, ↓ cognitive decline	[70]
Curcumin	HFD-fed mice	Curcumin alters gut microbiota and BA metabolism (↑ DCA, LCA)	TGR5 activation in thermogenic adipose tissue; effect lost in TGR5^−^/^−^ mice	↓ body weight, ↑ thermogenesis, ↑ UCP1	[71]
	ob/ob mice	Curcumin increases L-cell population via gut–BA modulation (↓ Lactobacillus, ↑ DCA)	Acts as TGR5 agonist/FXR antagonist in intestine; ↑ GLP-1 via TGR5 signaling	↑ energy expenditure, ↓ glucose, ↑ GLP-1 secretion	[72]
	NASFL patients	Curcumin shifts gut microbiota (↑ Bacteroides), ↑ serum DCA levels	↑ TGR5 expression in PBMCs, ↑ GLP-1 secretion in serum after 24 weeks	↓ hepatic fat, ↓ body weight, ↓ insulin resistance, ↑ GLP-1	[73]
Betulinic Acid (BA)	In vitro (reporter assay & SAR study)	Identified BA as a selective TGR5 agonist over FXR; induces GLP-1 release	Activates TGR5 in intestinal cells; no FXR cross-activation	↑ GLP-1 secretion, potential appetite suppression	[22]
	Humanized TGR5(H88Y) mouse model	BA analogs improve glucose metabolism in species-specific human TGR5 model	Strong TGR5 activation in human TGR5 knock-in mice; improves insulin sensitivity	↑ Glucose tolerance, ↑ insulin sensitivity	[76]
	Gut-restricted analog development	Synthesized BA analogs (e.g., 22-Na) with low systemic exposure to avoid gallbladder filling	Retained intestinal TGR5 activity while minimizing systemic TGR5 activation	↑ Local TGR5 effects without systemic side effects (gallbladder risk)	[77]
Ursolic acid (UA)	Type 1-like diabetic rats	UA increases cAMP in enteroendocrine L-cells	Activates intestinal TGR5 → GLP-1 secretion ↑	↑ GLP-1 → enhanced insulin secretion and glucose control	[78]
5β-scymnol	HEK293 cells (TGR5 overexpression)	↑ intracellular [Ca^2+^] via Gαq pathway (blocked by UBO-QIC)	Strong and sustained TGR5-specific activation by marine bile compounds	Potential activation of energy-related TGR5 pathways	[79]
Quinovic acid (QA) and derivatives	STC-1 intestinal L cells	Activation of TGR5 signaling; increased expression of proglucagon, PC1/3, and GIP	Direct TGR5 agonist activity; enhanced GLP-1 biosynthesis and secretion	Potential anti-obesity effect via incretin pathway activation	[80]
Obacunone	Obese KKAy mice, in vitro reporter assays	↑ TGR5 and PPARγ transcriptional activity; ↑ TGR5 mRNA in muscle	Acts as TGR5 agonist and upregulates its expression in muscle	↓ body weight, ↓ adiposity, ↑ muscle mass, ↓ blood glucose	[81]
Nomilin	HFD-fed mice, TGR5 luciferase reporter assay	Activates TGR5 and downstream cAMP signaling	Confirmed TGR5 agonist via luciferase activity	↓ body weight gain, ↓ fasting glucose	[59]

Abbreviations: CaN: Calcineurin, HFD: High-fat diet, AC: Adenylyl cyclase, MEF2: Myocyte enhancer factor 2, PGC-1α: Peroxisome proliferator-activated receptor gamma coactivator 1-alpha, DIO: Diet-induced obesity, BA: Bile acids, DCA/LCA: Deoxycholic acid/Lithocholic acid, GLP-1: Glucagon-like peptide-1, PBMCs: Peripheral blood mononuclear cells, FXR: Farnesoid X receptor, UCP1: Uncoupling protein 1, NASFL: Nonalcoholic simple fatty liver, GLP-1: Glucagon-like peptide-1, FXR: Farnesoid X receptor, SAR: Structure–activity relationship.

**Table 4 biomedicines-13-02405-t004:** Molecular docking results of selected natural products against the human TGR5 receptor (PDB ID: 7BW0). Natural products were selected based on known or predicted bile acid receptor activity and downloaded in SDF format from PubChem. The crystal structure of the human TGR5–Gαs complex (PDB ID: 7BW0) was used as the receptor structure. Molecular docking was performed using the CB-Dock2 server, which identifies potential binding cavities and executes blind docking based on AutoDock Vina scoring. Binding affinities (kcal/mol), interaction types (hydrogen bonds, van der Waals, π–π, and alkyl interactions), target residues, and predicted docking sites (orthosteric or shallow orthosteric) are summarized for each compound.

Compound	Structure	BindingScore(Kcal/mol)	Residual Target	DockingSite
Oleanolic acid(OA)	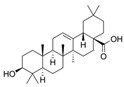 C_30_H_48_O_3_	−8.8	▪ Conventional hydrogen bonds:Leu71, Leu74▪ van der Waals interactions:Asn93 (N93^3.33^), Ser247 (S247^6.58^), Tyr89, Pro92, Trp75, Val88, Leu166, Val170, Val248, Tyr167, Tyr251	Orthosteric site
Curcumin	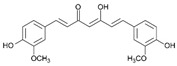 C_21_H_20_O_6_	−8.8	▪ Hydrogen bonds:Asn93 (N93^3.33^), Ser247 (S247^6.58^)▪ Hydrophobic interactions (Alkyl and π–Alkyl):Leu246, Leu265, Leu266, Leu71	Orthosteric site
Betulinic Acid(BA)	 C_30_H_48_O_3_	−7.3	▪ Hydrogen bond:Ser156 (S156), Ser157 (S157)▪ Pi–alkyl interaction:Trp149 (W149)▪ van der Waals interactions:Asn147 (N147), His148 (H148), Cys155 (C155), Tyr251 (Y251), Glu252 (E252)	Shallow orthosteric site
Ursolic acid (UA)	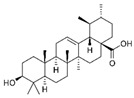 C_30_H_48_O_3_	−9.0	▪ Hydrogen bond:Val248 (V248), Trp75 (W75)▪ Unfavorable acceptor–acceptor interaction:Leu74 (L74)▪ van der Waals interactions:Leu71 (L71), Tyr89 (Y89)	Shallow orthosteric site
5β-scymnol	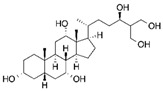 C_27_H_48_O_6_	−8.7	▪ Hydrogen bond:Asn93 (N93^3.33^), Ser247 (S247^6.58^), Tyr251 (Y251^7.31^), Leu244 L244^6.55^), Val248 (V248^6.59^)▪ Pi–alkyl interactions:Tyr89 (Y89^2.61^), Trp75 (W75^1.55^)▪ Van der Waals interactions:Phe96 (F96^3.36^), Leu166 (L166^5.40^)	Shallow orthosteric site
Quinovic acid (QA)	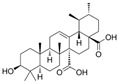 C_30_H_46_O_5_	−9.1	▪ Hydrogen bond:Asn93 (N93^3.33^), Leu74 (L74^2.66^)▪ van der Waals interactions:Ser157 (S157^4.57^), Phe96 (F96^3.36^), Tyr251 (Y251^7.31^), Leu166 (L166^5.40^)	Shallow orthosteric site
Obacunone	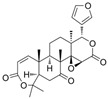 C_26_H_30_O_7_	−9.9	▪ Hydrogen bond:Asn93 (N93^3.33^), Tyr251 (Y251^7.31^)▪ Pi–alkyl interaction:Leu71 (L71^1.49^)▪ van der Waals interactions:Ser247 (S247^6.58^), Ser157 (S157^4.57^), Leu166 (L166^5.40^), Leu74 (L74^2.66^), Tyr89 (Y89^2.61^)	Shallow orthosteric site
Nomilin	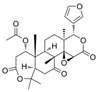 C_28_H_34_O_9_	−9.7	▪ Hydrogen bond:Asn93 (N93^3.33^)▪ Pi–alkyl interaction:Leu71 (L71^1.49^)▪ Pi–pi T-shaped interaction:Phe96 (F96^3.36^)▪ van der Waals interactions:Leu166, Leu244, Leu266, Val170, Val248, Ser247, Ser270	Shallow orthosteric site

## Data Availability

The original contributions presented in this study are included in the article. Further inquiries can be directed to the corresponding author(s).

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
