# Peer review of "Structure-Based Insights into TGR5 Activation by Natural Compounds: Therapeutic Implications and Emerging Strategies for Obesity Management"

_biomedicines, 2025, doi:10.3390/biomedicines13102405_

Round 1

Reviewer 1 Report

Comments and Suggestions for Authors

This review provides a thorough examination of the bile acid receptor TGR5 as a therapeutic target in obesity and related metabolic disorders. It discusses the receptor’s structural biology, signaling pathways, and ligand-binding features, with particular attention to orthosteric and allosteric modulation. A significant portion of the paper is dedicated to natural products with TGR5 agonist activity and their potential metabolic benefits. The docking analyses add a valuable layer of mechanistic insight, illustrating how different compounds interact with key residues and supporting the idea of ligand-specific activation. The authors also highlight future opportunities, including structure-guided drug design, gut-restricted ligands, and biased agonists, all aimed at enhancing therapeutic efficacy while limiting systemic side effects. Taken together, the review integrates metabolic, pharmacological, and structural perspectives to propose a framework for advancing TGR5-based therapies against obesity and related conditions.

The strengths of the manuscript include its wide-ranging scope, covering everything from receptor structure and signaling mechanisms to natural compound pharmacology and docking studies. The integration of natural products is particularly valuable, as most existing reviews focus primarily on synthetic ligands. Another strength lies in the mechanistic depth, with clear connections drawn between molecular interactions and systemic effects such as thermogenesis and glucose regulation. The docking component is also a strong addition, as it provides structural explanations for ligand binding preferences and selectivity.

That said, the paper has some weaknesses. Certain points, particularly GLP-1 secretion and thermogenesis, are repeated across multiple sections and could be streamlined. The review places heavy emphasis on preclinical and in vitro findings but gives limited attention to clinical evidence, which would help contextualize the therapeutic relevance of these compounds. Docking results are presented with strong confidence, but there is little discussion of their limitations as predictive tools. In addition, the review often leans toward highlighting positive findings without fully addressing adverse effects or practical challenges.

A few additional points warrant clarification. The relationship between shallow orthosteric binding and partial or biased agonism is mentioned but not clearly explained, which could be difficult for readers who are less familiar with GPCR signaling. Species-specific differences in bile acid derivatives are acknowledged but not sufficiently explored in terms of translational implications. While the idea of allosteric modulation is raised, the review does not provide concrete examples of natural products acting in this way, making the section feel more speculative than evidence-based.

To strengthen the paper, the authors could expand the clinical context by indicating which natural compounds have been tested in humans and identifying areas where data are still lacking. A short discussion of docking caveats would help temper overinterpretation of computational findings. Finally, offering a more critical appraisal of practical challenges, such as poor bioavailability, side effects, and variability in plant extract preparations, would give the review a more balanced and realistic perspective.

Author Response

  1. Streamline overlapping content by reducing repeated descriptions of GLP-1 secretion and thermogenesis.

⟶ Redundant descriptions of GLP-1 secretion and thermogenesis were removed, with detailed mechanisms retained in Sections 2.1 and 2.3 and later references streamlined through cross-references.

  1. Expand the clinical context by including natural compounds tested in humans and highlighting areas where clinical data are lacking.

⟶The revised manuscript includes additional content highlighting curcumin as a natural compound with clinical data, noting the absence of human studies for other natural products, and emphasizing the existing clinical gap.

  1. Add a discussion of docking limitations to avoid overinterpretation of computational findings.

⟶ A statement on the inherent limitations of docking has been added in Section 5, noting that results are hypothesis-generating and should be cautiously interpreted with experimental validation.

  1. Provide a more balanced view by addressing adverse effects and practical challenges such as poor bioavailability and variability of plant extracts.

⟶ It has been added to Section 7 (Discussion).

  1. Clarify the relationship between shallow orthosteric binding and partial or biased agonism for readers less familiar with GPCR signaling.

⟶ An explicit explanation of how shallow orthosteric binding may lead to partial or biased agonism through weaker toggle switch engagement and limited TM6 displacement has been added to Section 3.1 for clarity.

  1. Elaborate on species-specific differences in bile acid derivatives and their translational implications.

⟶ Section 4.3 has been revised to elaborate on the species-specific differences of BA derivatives, noting the weaker activity in murine TGR5 compared with human/canine receptors and emphasizing the importance of humanized models for translational research.

  1. Reframe the section on allosteric modulation by noting the lack of concrete natural product examples and treating it as a future research opportunity.

⟶  Section 6.2 has been revised to clarify that no natural products have been experimentally confirmed as TGR5 allosteric modulators and to frame allosteric modulation as a future research opportunity rather than established evidence.

Reviewer 2 Report

Comments and Suggestions for Authors

This manuscript provides a timely, comprehensive, and well-structured review on the structural mechanisms of TGR5 activation and the therapeutic potential of natural compounds in obesity management. The integration of recent high-resolution structural data (e.g., from cryo-EM studies) with molecular docking analyses of natural agonists represents a significant strength and adds original value to the work. The authors effectively summarize complex signaling pathways and clearly articulate emerging therapeutic strategies, such as gut-restricted agonism and biased signaling. The manuscript is logically organized, supported by an extensive and up-to-date bibliography, and makes a valuable contribution to the field. It is suitable for publication after minor revisions addressed below.

  1. Figure 2 Correction: The label "Gs binding interface" in Figure 2 should be corrected to "Gαs protein-binding interface" for precise biochemical terminology consistent with the text.
  2. On line 179, the reference to "compound 6" should be briefly contextualized or its chemical nature/origin indicated  (e.g., "the synthetic agonist compound 6" or citing the relevant reference, Macchiarulo et al., 2013) to enhance clarity for readers not intimately familiar with the primary literature.
  3. The molecular docking section (Section 5) would benefit from a brief sentence noting the validation status of the CB-Dock2 method or its prior application to GPCRs like TGR5, which would further strengthen the confidence in the presented interaction profiles.

Author Response

1. Figure 2 Correction: The label "Gs binding interface" in Figure 2 should be corrected to "Gαs protein-binding interface" for precise biochemical terminology consistent with the text.

⟶ The label in Figure 2 has been corrected from "Gs binding interface" to "Gαs protein-binding interface" for precision and consistency with the text.

2. On line 179, the reference to "compound 6" should be briefly contextualized or its chemical nature/origin indicated  (e.g., "the synthetic agonist compound 6" or citing the relevant reference, Macchiarulo et al., 2013) to enhance clarity for readers not intimately familiar with the primary literature.

⟶ The description has been revised to specify that compound 6 is a synthetic bile amine agonist reported by Macchiarulo et al. [65], providing clearer context for readers unfamiliar with the original literature.

3. The molecular docking section (Section 5) would benefit from a brief sentence noting the validation status of the CB-Dock2 method or its prior application to GPCRs like TGR5, which would further strengthen the confidence in the presented interaction profiles.

⟶ A sentence was added in Section 5 noting that CB-Dock2 has been benchmarked across diverse protein–ligand systems with ~85% success rate in binding pose prediction (RMSD < 2.0 Å), supporting its suitability for generating interaction hypotheses.